# Outreach Efforts to Prevent Newcastle Disease Outbreaks in Southern California

**DOI:** 10.3390/v14071509

**Published:** 2022-07-09

**Authors:** Alejandra Figueroa, Esteban Escobedo, Marco Solis, Charlene Rivera, Ann Ikelman, Rodrigo A. Gallardo

**Affiliations:** 1Department of Population Health and Reproduction, School of Veterinary Medicine, University of California, Davis, CA 95616, USA; alefigueroa@ucdavis.edu; 2California Department of Food & Agriculture, Ontario, CA 95814, USA; esteban.escobedo@cdfa.ca.gov (E.E.); marco.solis@cdfa.ca.gov (M.S.); charlene.rivera@cdfa.ca.gov (C.R.); ann.ikelman@cdfa.ca.gov (A.I.)

**Keywords:** Newcastle disease virus, biosecurity, gamefowl, poultry, outreach, education

## Abstract

In May 2018, an outbreak of virulent Newcastle disease (vND) affected backyard and commercial premises in Southern California. The re-occurrence of these outbreaks since the 1970s suggests that some poultry communities may not have reliable and stable resources available regarding biosecurity and disease prevention. Therefore, staff at the University of California, Davis (UCD) School of Veterinary Medicine (SVM), and the California Department of Food and Agriculture (CDFA) began organizing educational events and learning more about the gamefowl breeder community through a needs assessment exercise, during which local feedstores and neighborhoods were also visited. Focus groups were organized with breeders in various cities within the regional quarantine area, established by the CDFA during the vND outbreak. The focus groups were aimed at creating open communication networks with gamefowl breeders in the affected area, as well as to learn about their current sources of information, learning preferences, and current management practices. With the input from gamefowl breeders, as well as funding and input from the CDFA and the United States Department of Agriculture (USDA), a quality assurance program called the “Gamefowl Wellness Program” was established. Educational content was created and published through the UCD Gamefowl Wellness Program poultry health website. Additionally, with the help of the CDFA, the USDA, and pharmaceutical companies, Newcastle disease vaccines and training for their application were provided to feedstores with gamefowl breeder clientele. Nurturing trust with these poultry communities allowed us to receive the information needed to develop effective outreach strategies that could better serve them. Responding to community concerns might be a way to garner the trust necessary to prevent or at least promptly detect foreign animal disease outbreaks.

## 1. Introduction

The vND virus has periodically had large outbreaks in California—in 1971–1974, 2002–2003, and most recently in 2018–2020—creating a dire need for educational outreach and extension services for gamefowl owners. Interestingly, the 2018–2020 vND outbreak affected poultry owners in the same Southern California counties as the previous 2002–2003 oubreak, further supporting the concept that individuals may not have or are not receiving the educational resources they need to uphold the health of their birds [1]. Considering the concentrated numbers of backyard birds that can be present in both suburban and urban areas [2], the 2018–2020 outbreak ultimately affected 476 premises in total. Among the affected areas, four were commercial premises [3]. These facts reinforce the importance of educating poultry owners as one of the ways of protecting the commercial poultry industry and, consequently, food security and human health.

Disease risk may be reduced by personalizing outreach efforts to fit the needs of specific poultry communities. For example, Thailand adapted its Highly Pathogenic Avian Influenza (HPAI) surveillance program to specific poultry systems present in the country through passive and active surveillance and better communication with veterinary officers, thus successfully controlling an HPAI outbreak in 2008. The Thai gamefowl community was specifically targeted for these efforts, but was resistant to recommendations and disease reporting requests made by veterinary authorities, leading policymakers to evaluate the structure and implementation of their disease surveillance programs [4]. The gamefowl community in Southern California has been associated with practices that, even though not fully confirmed, might be introducing foreign animal diseases to the state and country. These communities have a complex social, ethnic, and legal background, and their practices have been established for years, regardless of the authorities’ directives. This creates the need for training and educating these communities on recognizing transmission pathways, reducing disease risk, and stimulating disease reporting, with the ultimate goal of guiding them to minimize disease introductions and outbreaks.

As with many gamefowl communities worldwide, owners of flocks at risk in California have demonstrated the same resistance to governmental authorities, while they have trust-based relationships among themselves, which are important in diseminating information [4,5]. Analyses of gamefowl networks in Thailand have shown that information is quickly diseminated between gamefowl owners and between different villages, primarily due to the efforts of gamefowl owners. Similar behaviors to those in Southern California, such as reluctance to report suspect cases, frequent movement of birds, free-ranging birds, and a general distrust of veterinary authorities, have been detected in gamefowl communities in Thailand and other countries [4,5]. Thus, outreach strategies may be more effective if they are re-evaluated and structured for the specific poultry communities that are being targeted, i.e., gamefowl poultry owners, with the objective of providing the resources they need in a format they can use. By conducting a primary needs assessment, followed by focus groups, we were able to learn about the gamefowl community in California, as well as their practices, needs, and beliefs, and using the collected information, after consultation with the authorities, the program was established.

## 2. Materials and Methods

### 2.1. Needs Assessment Period

To understand the structure and organization of gamefowl owners, a needs assessment of gamefowl communities was conducted, focusing on gamefowl breeders and flocks at risk in Southern California, from February to July 2019. Visits were made to neighborhoods and feedstores located within the CDFA regional quarantine area so as to speak with and learn from community leaders about how they were being impacted by the vND outbreak at the time. Visits did not consider flock or bird assesments, and a strict protocol of biosecurity was implemented for veterinarians and technicians visiting the community.

To address the community needs for education, three in-person educational workshops for backyard poultry enthusiasts and gamefowl breeders were offered. At these workshops, staff from the Western University of Health Sciences, the CDFA, and the UCD SVM presented on topics related with poultry disease prevention, biosecurity, vaccination, and updates on the vND outbreak. In addition, information about the educational needs of the attendees was recorded. One workshop was conducted in the city of Hesperia, and the other two at Western University of Health Sciences. To reach a wider audience during this outreach and needs assessment period, an online presence was established by reporting information regarding the vND outbreak in English and Spanish on the UCD SVM’s Facebook and Twitter social media pages. The attendance numbers of the workshops encouraged us to try a different approach in the form of focus groups, with community leaders, in order to learn how to better reach gamefowl communities.

### 2.2. Focus Groups

From August 2019 to January 2020, five focus groups were conducted with gamefowl breeders within the vND regional quarantine area in the cities of Perris, Compton, and Bloomington. Focus group participants were recruited through network connections with CDFA staff, as well as by speaking with feedstore clientele and owners. The groups served not only to provide educational resources to breeders, but to learn about the at-risk gamefowl communities. Gamefowl breeders were asked a series of questions at the focus groups, including active organizations or groups they belonged to, sources of gamefowl information and products, topics on poultry health they wanted to learn about, format and language of preference for educational materials, health and management practices they utilized within their flocks, and who were the decision makers for their flocks. Staff from the UCD SVM, the CDFA, and the local California Animal Health and Food Safety (CAHFS) laboratory in San Bernardino listened to gamefowl breeders’ concerns and provided feedback. Additionally, breeders were taught where to find reliable sources of information to answer their questions, as well as how to properly send samples or dead birds to the CAHFS laboratory for diagnostic testing.

### 2.3. Gamefowl Wellness Program

Using the information obtained from the educational workshops and focus groups with gamefowl breeders, the objective became to bring resources directly to these communities in the form of a program, with the purpose of addressing education, and outreach for gamefowl flocks. The program was initiated in October 2020. Because of COVID-19 pandemic restrictions, free virtual workshops were provided on the topics that were previously recognized as a need during the needs assessments and focus groups: vaccination protocols, biosecurity, disease prevention, and diagnostic lab submissions. Participants were also given the opportunity to ask questions. During the first virtual workshop, attendees were polled on four topics: number of birds in their flock, how well they thought they practiced disease prevention, the vaccination status of their flock for NDV, and the source of their birds. In addition, local feedstores were visited and outreach events were organized. These feedstores were located in the cities of Riverside, Compton, El Monte, Hesperia, Fontana, Muscoy, Perris, and Bloomington. As part of the outreach effort with feedstores, Newcastle disease vaccines were also distributed. As opposed to commercially available vaccines (10,000 doses), 1000-dose vaccine vials were distributed. In addition, eye droppers were included to allow for individual application. These products were donated by Zoetis (Newcastle Disease Vaccine, B1 Type, LaSota strain, 10 Sylvan Way, Parsippany, NJ 07054, USA), and Boehringer Ingelheim (Newcastle Disease B1 Type La Sota strain, 3239 Satellite Blvd. NW, Duluth, GA 30096, USA).

Through the networking developed with community leaders and breeders, the program was promoted to a broader audience. Mainstream media and a podcast series were used as marketing and dissemination tools. Detailed information on vND and disease prevention was presented during three radio interviews with a local radio station in Southern California, which were also distributed online through the radio station’s Facebook social media page (https://www.facebook.com/LaRanchera967/ (accessed on 20 December 2021)). Additionally, seven podcast episodes on common poultry diseases and biosecurity were conducted and distributed (https://www.breedersacademy.com/?s=rodrigo+gallardo (accessed on 20 December 2021)).

To help diseminate accurate information through the program, a Facebook social media page was established (https://www.facebook.com/ucdavis.poultry (accessed on 20 December 2021)), as well as a program email address (poultry@ucdavis.edu (accessed on 20 December 2021)) through which gamefowl breeders could reach out to poultry experts with their concerns in either English or Spanish. A poultry health webpage (https://www.vetmed.ucdavis.edu/poultry-health (accessed on 20 December 2021)) was created where podcast interviews, informational factsheets, instructional videos, and workshop information in English and Spanish were periodically posted.

## 3. Results

### 3.1. Needs Assessment Period

Through interaction with gamefowl breeders, a reluctance to report disease outbreaks and to work with diagnostic laboratories was detected. Most breeders in the area did not trust governmental agencies and the information they provided. Several people suggested hosting educational workshops in places where gamefowl breeders woud feel comfortable and they could interact and establish trust relationships. Therefore, three in-person workshops were hosted. Considering the low workshop attendance, i.e., 15, 5, and 5, respectively, efforts were made to establish an online presence for outreach purposes. However, the poor interaction from the gamefowl community was reflected online. The SVM’s 2 Facebook posts on vND garnered only 3 “likes”, while the 4 posts on Twitter collectively garnered 17 “retweets”. This poor response reflected a failure in our interactions with our target population of poultry breeders. As a response, focus groups were conducted to really understand the group needs.

### 3.2. Focus Groups

Five to six gamefowl breeders attended each of the four focus groups hosted in Compton and Bloomington. In Perris, the interest was high, with 23 attendees. Through the focus groups, important information regarding gamefowl activities was collected. There are no active breeder organizations or clubs within their communities, their most important source of information is through non-professional poultry enthusiasts (friends and neighbors), the internet, veterinarians and breeders in Central America, and breeders in the U.S. The products used for disease prevention and treatment, i.e., vaccines, antibiotics, vitamins, etc., are obtained from outside the U.S. and often have labels in foreign languages. Additionally, they reported using veterinary products without a clear understanding of their purpose or how to use them. These products are chosen based on feedstore owner recommendations and affordability. In terms of NDV vaccines, gamefowl breeders reported reconstituting 10,000-dose vaccine vials and vaccination via drinking water.

Some of the topics expressed as educational needs were nutrition, management, welfare, and disease prevention. They preferred short videos and factsheets in English and Spanish.

It was also evident that decision makers in the gamefowl breeder community were males, usually 50 years old or older, creating the need to brainstorm outreach strategies for older communities in order to design outreach materials useful to them.

### 3.3. Gamefowl Wellness Program

Virtual workshops were conducted to safely engage with gamefowl breeders during the COVID-19 pandemic. Twenty-seven gamefowl breeders attended the first online workshop. A brief description of the program was presented along with a disease prevention lecture. These two presentations were followed by a question and answer session. After the meeting, attendees were polled on four topics: (i) number of birds in their flock, (ii) how well they thought they practiced disease prevention, (iii) NDV vaccination status of their flock, and (iv) source of their birds. Not all workshop participants answered the poll questions and some received more responses than others. Differing flock sizes were reported and the most frequent size among the attendees was between 51 and 100 birds (Table 1). The second most frequent flock size was between 0 and 50 birds, and only 2 breeders reported having more than 100 birds.

While most of the breeders (8/12) responded that they applied disease prevention methods as “moderately good or good”, only 2 stated they did it “poorly” and 2 qualified themselves as “excellent” (Table 1). Most of the attendees (7/15) reported vaccination against NDV and 3/15 needed additional information to decide (Table 1). Finally, birds were obtained in and out of the state, while none of the attendees disclosed purchasing birds out of the country (Table 1).

The second virtual workshop hosted 18 gamefowl breeders and 35 viewers joined through the program’s social media page. The workshop was recorded and published reaching 9245 views, 478 “engagements”, and 89 shares. Facebook engagements are characterized as the overall number of interactions that users have with a post, such as likes, shares, and clicks on links; the metric demonstrates how pertinent the post is to the audience [6]. While no evaluations on workshop quality or usefulness were offered to participants during the first workshop, feedback from the second workshop was overall positive in topics such as quality of information presented, visual aids, knowledge of subject, and presentation skills. Presentations were rated as “excellent”. Additionally, participants stated they were “extremely likely” to attend another workshop and that they found the covered material “extremely useful” and “very useful”.

In terms of the program’s online presence, the Facebook social media page currently had 1091 followers and the poultry health webpage had over 2300 visits since its establishment in September 2020. Sixty-six emails were received seeking advice for sick or injured birds, information on other gallinaceous species and poultry diseases, on diagnostic services for their flock, and on how to join the program.

In-person visits to feedstores were performed to promote the program and to provide educational materials. Feedstores that were within the regional quarantine area were targeted (Figure 1). Over 168,000 vaccine doses were distributed to 11 feedstores along with factsheets on correct vaccination protocols, procedures, and schedules. Sign-up sheets for the program were left with feedstore owners so their clientele could provide their information to join.

So far, the prgram enrolled 135 participants through feed stores, outreach events, online, and CDFA contacts. Figure 2 shows the cities where the program paricipants resided. Most of the enrolled participants lived near Riverside and Los Angeles. Interestingly, a higher concentration resided in the Los Angeles area as opposed to the San Bernardino area (Figure 2).

## 4. Discussion

Disease outbreaks of backyard poultry and gamefowl pose a significant risk to the commercial poultry industry. A 2007 survey of gamefowl breeders and backyard poultry owners in the U.S. revealed that gamefowl breeders had more frequent movement of birds on and off their premises than backyard owners [7]. An example is the 2018–2020 vND outbreak, when, despite the California State Veterinarian bird movement restriction, gamefowl were transported out of the qurantine region [8]. These movement has been previously associated with difficulties in the control of foreign animal diseases [8]. Additionally, gamefowl breeders reported more flock health problems compared with backyard flock owners, even though gamefowl breeders reported vaccinating their birds [7]. This suggests that vaccination and biosecurity practices are not being properly utilized by some gamefowl breeders. Furthermore, gamefowl breeders can remain established within a region for 20 or more years [7]. Such information stresses how crucial it is for veterinary professionals and extension specialists to create and nurture communication networks with these gamefowl communities.

In-person workshops on poultry health topics were conducted along with attempts to establish an online presence for outreach purposes. Unfortunately, workshop attendance was low, and some attendees mentioned this was due to mistrust in government authorities. Research shows that this mistrust is widespread and is displayed in gamefowl communities worldwide. A low level of trust in agricultural department offices or even veterinarians has been shown as a critical reason for under-reporting disease outbreaks in Australia [9], the U.K. [10], The Netherlands [11], France [12], and in Asian countries [13]. Gamefowl breeders avoid reporting to avoid a negative emotional impact, moral fault, and loss of their valued breeds [5].

During our initial interactions with gamefowl breeders, we detected features that reflected special outreach needs. Common outreach performed through workshops or general social media information did not work at this time, as individuals did not feel comfortable outside their communities or visiting webpages associated with institutions they did not know, i.e., the UCD. A trust relationship was needed in order to understand the needs of the gamefowl community, outreach format, and online promotion. For this, community leaders have been instrumental in relating to these breeders.

Through focus groups, we established long lasting relations explaining the program role and how they could benefit from it, and we understood the problems these breeders face mainly in disease prevention and management. As far as disease prevention practices, the use of antibiotics, vaccines, and other veterinary products is widespread, without a clear understanding of what they are being used for or how. Breeders reported obtaining their veterinary products from outside the country with instructions in foreign languages, which could cause serious confusion between the active compounds and their use. For example, they might refer to an antibiotic or vitamin as a vaccine. A large percentage of gamefowl breeders use vitamins and antibiotics to treat disease in their flocks [14], and antibiotic use in food animals has been reported to be “five times higher” in the U.S than in the U.K. [15]. Breeders also reported using vaccines meant for commercial poultry use. While large-dose vaccines are useful in the commercial industry, they are difficult to handle, reconstitute, and apply in small flocks; if not reconstituted adequately, combined viruses in vaccine vials, i.e., infectious bronchitis virus and NDV, if improperly diluted, can compete for the receptors in the cells and induce poor protection in these flocks [16]. All of this results in poor immunization and missuse and misstrust of certain products. For these reasons, it is imperative to focus on education in disease prevention in order to reduce the need for using medications. This situation highlights the need for outreach, not only for poultry breeders, but also for feedstore owners, so they can help promote a judicious use of antibiotics, biologics, and other veterinary products.

Internet and social media might not be the right choice for outreach, if we consider that a high percentage of breeders do not feel comfortable with these communication platforms. This illustrates the need for trustworthy and properly catered sources of knowledge. Even with advice from others, breeders stated they would only trust their own methods, as they believed that they were more knowledgeable and experienced than the veterinary services provided to them [4]. Such beliefs highlight another issue: the lack of veterinarians trained in poultry medicine and the mistrust these communities have in the veterinary services provided locally.

Anonymous polls were conducted during virtual workshops offered by the Gamefowl Wellness Program. The reported flock sizes (Table 1) differed from previously published information stating that at a national basis, a high percentage of gamefowl breeders (64.6%) owned 100 birds or more [14]. However, it is likely that participants who attended the program’s virtual workshop may not have been completely honest about their flock sizes because they felt they could not be as anonymous as they wanted to be. In addition, the data collected were from a virtual workshop that was provided to individuals residing in urban areas of Southern California, where there may not be a lot of space for them to have large flocks. While some workshop attendees did provide valuable information and insight, this sample was probably not representative of the gamefowl owners across the country. In addition, minimal data on general backyard flock size or health within the State of California exist [17]. Thus, additional studies that collect similar information from gamefowl communities are needed.

In contrast with what the gamefowl owners stated in terms of disease prevention proficiency (Table 1), according to a national survey, gamefowl breeders (and backyard flock owners) focussed on washing their hands only after handling their birds, instead of both before and after. They also frequently took their birds to external locations where other birds were present and subsequently returned them to their flock without a quarantine period [14], increasing the risk of transmitting infectious pathogens to their flock. Nevertheless, considering the value of their birds, gamefowl breeders engage in good biosecurity measures that are not commonly used by other poultry enthusiasts. For example, 96.5% of gamefowl breeders in the U.S. reported separating new birds from their flock [14].

Most of the poll responders in the virtual workshop (7/15) were vaccinating against Newcastle disease (Table 1). This is consistent with a USDA report of gamefowl breeders throughout the U.S., which demonstrated that more than half of breeders polled (58.6%) had vaccinated their birds within the last 12 months [14]. Testing vaccine takes and outreach on correct vaccine application is needed to make sure vaccinations are translated into the immunization of these flocks. It is common to find breeders that do not understand the concept of a vaccination program or the importance of periodical boosters. Lastly, most of the breeders polled (10/15) stated they purchased birds from out of California. (Table 1). This did not completely coincide with a national survey of breeders, which found that only 38.1% of respondents purchased birds from outside their state [14]. However, one must consider that out-of-state purchases could depend on the geographical location of the breeder; some states may have more availability of desirable gamefowl than others, leading more Californian breeders to purchase out-of-state birds. In terms of international bird purchases, our data were consistent with other findings showing that not many gamefowl breeders purchased birds from outside the country [14].

Social networking of cockfighting communities in Thailand allows for information on disease to be shared rapidly both within the same village and neighboring villages, to reduce the risk of disease spread [5]. Thus, efforts were aimed at providing educational content to the individuals deemed trustworthy by these communities. Collaborations with local radio and through podcasts produced by people recognized by gamefowl enthusiasts helped the program gain followers on Facebook and increased email communications from individuals wanting to learn about poultry health. Gamefowl breeders have been described as composing a “community of practice” that relies on strong, internal bonds and commitment to each other [4]. Individuals involved with such communities must first establish a reputation of trustworthiness in order to be a part of it [18], further supporting the idea of feedstore owners as the ideal candidates to pass along information and requested supplies, such as the Newcastle disease vaccines. By educating feedstore workers and owners, outreach specialists may be more effective at diseminating educational content to target communities, while simultaneously supporting their self-sufficiency.

Most of the program’s enrolled participants lived in Los Angeles. This is one of the weaknesses of the program and it reflects the need for more resources and expertise in order to reach other areas in Southern California. This finding was consistent with a CAHFS study that analyzed data on their poultry submissions, but even so, the actual number of gamefowl breeders within city limits may be higher [17]. The close proximity between flocks in urban neighborhoods is concerning; it can contribute to disease transmission, which could potentially affect the commercial poultry industry. Backyard flocks close to commercial poultry facilities have previously demonstrated the presence of antibodies against numerous pathogens via serological tests, some of which include Newcastle disease, *Salmonella pullorum*, avian encephalomyelitis, *Bordetella avium*, hemorrhagic enteritis, infectious bronchitis, and infectious bursal disease [19]. As the challenge in recruitment for the program was likely due to mistrust and fear of government entities, in the future, the confidentiality of enrollment would need to be emphasized. As the program continues, the objective is to encourage breeders to use diagnostic laboratories on a regular basis and to practice preventative veterinary medicine, rather than to seek consultations when birds are sick.

## 5. Conclusions

Maintaining open communication networks with gamefowl breeders in Southern California has helped create the strong trust-based relationships necessary to provide them with the resources they have asked for. Distributing accurate information in several formats online could help reduce biosecurity risks [17]. However, electronic communication, while convenient, should not entirely replace personal contact [20]. By maintaining a constant presence in these communities, whether virtual or in-person, trust-based relationships can be nurtured to collaboratively contribute to overall avian health. Enhancing communication relies not only on the scientific rationale behind certain decisions, but also on individuals’ beliefs, values, and the level of trust they have in the sources of information [21]. Actively listening to the input of gamefowl breeders will allow us to maintain their trust, improve our outreach strategies, and learn how to respond more effectively to their concerns. Over time, these communication networks may help reduce problems affecting flock health, prevent future disease outbreaks, and, more importantly, encourage trust with the gamefowl breeder community, which can help continue this cycle.

## Figures and Tables

**Figure 1 viruses-14-01509-f001:**
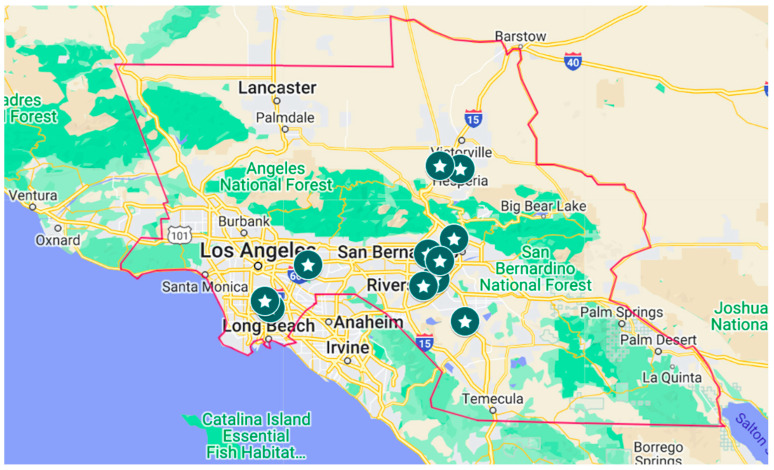
Feedstores visited (✩) for Newcastle vaccine distribution within the regional quarantine area (red line) that was established by the CDFA during the 2018–2020 vND outbreak in Southern California.

**Figure 2 viruses-14-01509-f002:**
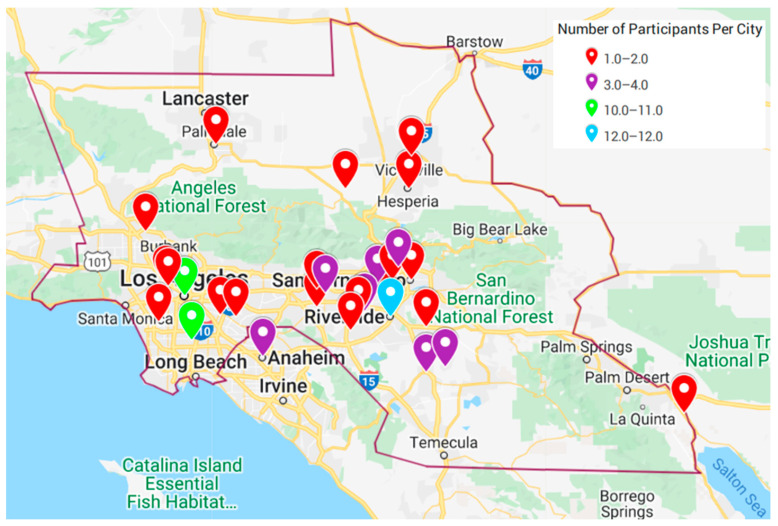
Residence cities of participants (pin) colored by number of participants enrolled in the Gamefowl Wellness Program. The red line denotes the CDFA regional quarantine area established during the 2018–2020 vND outbreak in Southern California.

**Table 1 viruses-14-01509-t001:** Virtual workshop poll answers from gamefowl attendees.

Poll Question	Number of Birds	Responses	Total Responses to This Question
**How many birds do you have in your flock?**	0–50	5	13
	51–100	6	
	101–500	2	
	More than 500	0	
**How well do you think you apply disease prevention methods in your flock?**	Poorly	2	12
	Moderately good	5	
	Good	3	
	Excellent	2	
**Do you vaccinate to protect your birds against Newcastle disease?**	Yes	7	15
	No	5	
	I want to learn more about vaccination first	3	
**Where do you purchase your birds?**	In-state (California)	5	15
	Out-of-state (in the U.S.)	7	
	Both in- and out-of-state	3	
	Abroad (overseas, other countries)	0	

## Data Availability

Not applicable.

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
