# Peer review of "Outreach Efforts to Prevent Newcastle Disease Outbreaks in Southern California"

_viruses, 2022, doi:10.3390/v14071509_

Round 1

Reviewer 1 Report

General Remarks:

This is an interesting article that looks at how community outreach and engagement can play an important role in disease response and prevention. However, the lack of integration between the actual response actions and the efforts described in this paper take away from the value of the paper. Additionally, the paper reads more as a desire of the authors to brag about their work, rather than a thoughtful look at one approach that was taken and how it might contribute to future efforts in disease control. I am hopeful that this tone can be addressed through strategic editing by the authors and through a more thorough literature review. There has been a lot of work down on risk communication, community outreach and engagement, and different studies have shown that not all outreach efforts are successful in leading to behavior change or adoption.

The low enrollment rate of the program and on-going concerns around confidentiality from this community suggest that further changes to the program may be needed. It would be helpful if the paper included more of a discussion on the pros/cons of this effort, the limitations of the program, including the costs, and clear ideas for next steps or how it might be evaluated in the future. Should the program continue? Is the work done and how would you know?

There are numerous grammatical errors in the paper, and the authors use different tenses at different times, which is confusing to the reader. Review and correction needed.

Abstract:

Line 9: Recommend removing the word “yet another” – this feels unnecessarily inflammatory. Since 1971, there have only been three outbreaks of vND in southern California, each of which was controlled through enormous effort and taxpayer dollars.

Line 12: “our team” – this sentence seems to imply that the work is solely from UC-Davis rather than a collaboration with CDFA. Who is “our team”? Suggest rewording.

Line 19: I am sure that the establishment of the Gamefowl Wellness Program was a much larger process, including the need for funding and federal support. Suggest rewording to highlight how this work contributed to the development of this program, one valuable piece of that entire process, while recognizing that there was a larger process involved.

Line 21: “our” - whose website?

Line 24-25 – verb tenses need to be addressed, is information continuing to be posted today?

Introduction

Line 43 “should” – suggest rewording, outreach is part of the answer, not THE answer - for example: Personalized outreach efforts designed to fit the needs of a community have been shown to help reduce disease risk.

Line 49 – would be helpful to know more about how they changed these programs to control disease and why they might still be struggling with endemic disease

Line 54 – again, training and education are a valuable part of the solution, not THE solution. The complex disease risk pathways associated with that region of the country would suggest that individual bird owner’s actions are unlikely to completely prevent the introduction or spread of disease. A holistic effort aimed at reducing risk, addressing loose birds and other transmission pathways, as well as supporting early reporting would be needed to fully address this risk. The authors may benefit from reviewing the CEAH epidemiologic reports released for this outbreak: Epidemiologic Analyses of Virulent Newcastle Disease in Poultry in California (usda.gov)

Line 58: poultry networks in what countries? Is that info relevant to S. California? If so, why?

Line 61: “must” suggest rewording. This is a scientific manuscript, which should focus on demonstrating the value, pros/cons, of a specific approach, without appearing to be close-minded or to already have all the answers. I would make this same suggestion for anywhere in the paper where must or should are used.

Line 63: what is sufficient knowledge? How is that known or measured? Suggest rewording – what did you really gain – vocabulary, an understanding of how they think about risk, a list of products they would like to receive, etc?

Line 66: How do you know that they were effective? What follow up was done to actually determine the impacts of disease risk?

Materials and Methods

Line 70-74: would suggest including an efforts made by the team to minimize their own risk of spreading disease. What biosecurity practices were used by the team to do this work in the middle of an outbreak?

Line 82: what information? Was the needs assessment a failure and focus groups were needed? How were needs being identified during these discussions? They just sound like community outreach sessions, not a needs assessment. What does a social media presence have to do with this at all?

Line 86: how were participants for these groups identified? What sort of bias might result from the way that participants were identified?

Line 93: “What were their community needs” – what is this referring to? Any needs (mental, social, healthcare, crime)? What community?

Line 100: Suggest moving this section to results and spending a little more time explaining how the initial focus groups contributed useful ideas and information for the development of the wellness program, recognizing that it was a much larger effort to create and fund the program. Also, it would be helpful to include at the start of this paragraph information about the wellness program, including its objectives and funding.

Line 109: Why are these vaccines any different than what they had been using during the outbreak?

Line 115: “we have been” – again bad word tense that needs to addressed. Is this work on-going?

Line 122: his webpage? Our program? And what poultry health webpage? – need to include links and clear website descriptors.

Results:

This section overall seems really sparse. It would be helpful to have a comprehensive summary of the outcomes of the need assessment.

Lines 142-161 – this section should be reworked. It feels like there is a blend between outcomes of the needs assessment, the authors’ interpretation of those needs, and then the actions the authors recommend based on the needs assessment. It would be helpful to split those out and treat each section more thoroughly. For example, the needs could be outlined here, and then the discussion would interpret those needs and align them with other outreach areas or recommendations from the literature that would support or clarify the need.

Line 162: The introduction of more virtual workshops here is confusing. If there was a survey, why was it not clearly outlined in the methods? Given that the survey data only represent 27 participants, for whom no demographic data are available, the results should not be presented as percentages. They are not likely to be representative of the population of gamefowl breeders in any way. I would recommend simply saying “11 out of the 27 workshop participants indicated…”

Line 175: one out of every 3 workshop participants were not vaccinating for ND despite the occurrence of an outbreak that had been going on for years. That seems pretty significant and not surprising that they might have questions around vaccination.

Maps – the map visuals were very fuzzy, would recommend using a higher quality map product for the publication

Discussion:

Line 235: when was this survey conducted? Was it relevant to your population of interest? If so, how?

Line 248: Reporting of suspect cases is a complex behavior, and research has shown that beliefs and perceptions may differ when an outbreak is known to be present (see Delgado et al, https://doi.org/10.1016/j.prevetmed.2014.09.01). It would be valuable to know if your engagement identified how perceptions might have changed due to the on-going outbreak.

Line 255: this feels like new material, which should have been included in the results section

References:

Link for reference 3 does not work and needs to be corrected

Author Response

Reviewer #1.

This is an interesting article that looks at how community outreach and engagement can play an important role in disease response and prevention. However, the lack of integration between the actual response actions and the efforts described in this paper take away from the value of the paper. Additionally, the paper reads more as a desire of the authors to brag about their work, rather than a thoughtful look at one approach that was taken and how it might contribute to future efforts in disease control. I am hopeful that this tone can be addressed through strategic editing by the authors and through a more thorough literature review. There has been a lot of work down on risk communication, community outreach and engagement, and different studies have shown that not all outreach efforts are successful in leading to behavior change or adoption.

The low enrollment rate of the program and on-going concerns around confidentiality from this community suggest that further changes to the program may be needed. It would be helpful if the paper included more of a discussion on the pros/cons of this effort, the limitations of the program, including the costs, and clear ideas for next steps or how it might be evaluated in the future. Should the program continue? Is the work done and how would you know?

We appreciate the reviewer’s critique; we think is valid. We reviewed and discussed the efforts of the program from a different and critical perspective throughout the complete manuscript. The program started 3 years ago and has evolved with the collected information and the engagement rates, we hope this version reflects the work we have done in the program’s establishment to improve its effectiveness.        

There are numerous grammatical errors in the paper, and the authors use different tenses at different times, which is confusing to the reader. Review and correction needed.

We addressed grammatical errors and reviewed the manuscript.

Abstract:

Line 9: Recommend removing the word “yet another” – this feels unnecessarily inflammatory. Since 1971, there have only been three outbreaks of vND in southern California, each of which was controlled through enormous effort and taxpayer dollars.

We followed the reviewer’s suggestions.

Line 12: “our team” – this sentence seems to imply that the work is solely from UC-Davis rather than a collaboration with CDFA. Who is “our team”? Suggest rewording.

This was re-worded

Line 19: I am sure that the establishment of the Gamefowl Wellness Program was a much larger process, including the need for funding and federal support. Suggest rewording to highlight how this work contributed to the development of this program, one valuable piece of that entire process, while recognizing that there was a larger process involved.

Addressed in lines 20 to 23.

Line 21: “our” - whose website?

This info was added in line 22

Line 24-25 – verb tenses need to be addressed, is information continuing to be posted today?

Verbal tenses were fixed

Introduction

Line 43 “should” – suggest rewording, outreach is part of the answer, not THE answer - for example: Personalized outreach efforts designed to fit the needs of a community have been shown to help reduce disease risk.

The sentence was re-worded.

Line 49 – would be helpful to know more about how they changed these programs to control disease and why they might still be struggling with endemic disease

The information was added in lines 104 to 106.

Line 54 – again, training and education are a valuable part of the solution, not THE solution. The complex disease risk pathways associated with that region of the country would suggest that individual bird owner’s actions are unlikely to completely prevent the introduction or spread of disease. A holistic effort aimed at reducing risk, addressing loose birds and other transmission pathways, as well as supporting early reporting would be needed to fully address this risk. The authors may benefit from reviewing the CEAH epidemiologic reports released for this outbreak: Epidemiologic Analyses of Virulent Newcastle Disease in Poultry in California (usda.gov)

These sentences were reviewed in lines 110 to 115.

Line 58: poultry networks in what countries? Is that info relevant to S. California? If so, why?

This was clarified in lines 121 to 123

Line 61: “must” suggest rewording. This is a scientific manuscript, which should focus on demonstrating the value, pros/cons, of a specific approach, without appearing to be close-minded or to already have all the answers. I would make this same suggestion for anywhere in the paper where must or should are used.

This was corrected here and elsewhere in the manuscript

Line 63: what is sufficient knowledge? How is that known or measured? Suggest rewording – what did you really gain – vocabulary, an understanding of how they think about risk, a list of products they would like to receive, etc?

This was addressed in lines 127 to 129. Specifics are covered in the results section of the manuscript.

Line 66: How do you know that they were effective? What follow up was done to actually determine the impacts of disease risk?

The sentence was re-worded, line 124.

Materials and Methods

Line 70-74: would suggest including an efforts made by the team to minimize their own risk of spreading disease. What biosecurity practices were used by the team to do this work in the middle of an outbreak?

A statement mentioning specifics about the visits and highlighting the biosecurity protocol used was added in lines 283 to 285.

Line 82: what information? Was the needs assessment a failure and focus groups were needed? How were needs being identified during these discussions? They just sound like community outreach sessions, not a needs assessment. What does a social media presence have to do with this at all?

Outreach sessions were also used to obtain information about their educational needs. This and social media presence comments were clarified in lines 290 to 297

Line 86: how were participants for these groups identified? What sort of bias might result from the way that participants were identified?

Participants were in the beginning identified by word of mouth, then we got help from social media and finally we detected community leaders. This information is stated in lines 279 to 296 and 576 to 577.

Line 93: “What were their community needs” – what is this referring to? Any needs (mental, social, healthcare, crime)? What community?

Community needs are reported throughout the results section. We are talking about Gamefowl breeders.

Line 100: Suggest moving this section to results and spending a little more time explaining how the initial focus groups contributed useful ideas and information for the development of the wellness program, recognizing that it was a much larger effort to create and fund the program. Also, it would be helpful to include at the start of this paragraph information about the wellness program, including its objectives and funding.

The establishment of the program and its resources were described in the materials and methods section. The response to the program was described in the results section. Objectives were included at the beginning of the paragraph and funding source is provided at the end of the manuscript.    

Line 109: Why are these vaccines any different than what they had been using during the outbreak?

Vaccines commercially available in Southern California are residues from commercial vaccine use in bottles of 10,000 doses. The vaccines distributed by the program were on 1,000 dose presentation with eye droppers.   

Line 115: “we have been” – again bad word tense that needs to addressed. Is this work on-going?

This was corrected

Line 122: his webpage? Our program? And what poultry health webpage? – need to include links and clear website descriptors.

This was summarized for clarity. Web addresses were provided.

Results:

This section overall seems really sparse. It would be helpful to have a comprehensive summary of the outcomes of the need assessment.

Lines 142-161 – this section should be reworked. It feels like there is a blend between outcomes of the needs assessment, the authors’ interpretation of those needs, and then the actions the authors recommend based on the needs assessment. It would be helpful to split those out and treat each section more thoroughly. For example, the needs could be outlined here, and then the discussion would interpret those needs and align them with other outreach areas or recommendations from the literature that would support or clarify the need.

The reviewer suggestions were followed. Important results were mentioned in an organized fashion and their relevance discussed on the discussion section.    

Line 162: The introduction of more virtual workshops here is confusing. If there was a survey, why was it not clearly outlined in the methods? Given that the survey data only represent 27 participants, for whom no demographic data are available, the results should not be presented as percentages. They are not likely to be representative of the population of gamefowl breeders in any way. I would recommend simply saying “11 out of the 27 workshop participants indicated…”

Virtual workshops and their surveys were introduced in the materials and methods section. Lines 597 to 603. Demographic information was not collected to protect the confidentiality of the responders. Results were reported as suggested.

Line 175: one out of every 3 workshop participants were not vaccinating for ND despite the occurrence of an outbreak that had been going on for years. That seems pretty significant and not surprising that they might have questions around vaccination.

We agree with the reviewer’s comment

Maps – the map visuals were very fuzzy, would recommend using a higher quality map product for the publication

Map quality was improved

Discussion:

Line 235: when was this survey conducted? Was it relevant to your population of interest? If so, how?

The survey is referenced in the manuscript and was performed in 2007. It is relevant to our work because involves small flocks and Gamefowl. See reference #7.

Line 248: Reporting of suspect cases is a complex behavior, and research has shown that beliefs and perceptions may differ when an outbreak is known to be present (see Delgado et al, https://doi.org/10.1016/j.prevetmed.2014.09.01). It would be valuable to know if your engagement identified how perceptions might have changed due to the on-going outbreak.

We agree with the reviewer comment. This topic is out of the scope of the present manuscript.

Line 255: this feels like new material, which should have been included in the results section

This sentence discusses the results obtained on the needs assessments; it is not new material.   

References:

Link for reference 3 does not work and needs to be corrected

This was done

Reviewer 2 Report

Figueroa et al. created the Gamefowl Wellness Program in response to the lack of education surrounding the 2018 Newcastle disease virus outbreak and prevention measures in California. Through a primary needs assessment, focus groups with breeders, and outreach with feedstores, the authors were able to determine the resources this community needed and in which formats they best responded to. This has allowed for preemptive work to be done in the communities that continue to see virulent Newcastle disease virus outbreaks.

In my opinion, this work has shown the gap between research completed in the laboratory and dissemination of that information into the community. Other projects like this one have been completed for other viral infections, but to my knowledge, this is the first for NDV, following the latest outbreak. The novelty and quality of this manuscript, along with the interest being there for a wide audience, I suggest that this article be accepted with minor revisions.

I do have a few minor remarks:

11.      Please post the links in the results section for the podcast and the facebook group, as these were part of the data for the article.

22.      Since questions are a part of the results for this study, a table may show this data in a format that is easier for others to replicate the study.

33.      Error bars need to be inserted in each figure that has a bar graph.

44.      Each graph should equal 100%. In one figure it equals 99%, one figure equals 101%, another 107%, and another 120%. Exact numbers (n=) would be useful in the results section since in some instances you only have 5 participants. Maybe with the table of questions?

55.      Did you have any control questions for this experiment? If so, please mention them. If not, this may be helpful in the future.

66.      Did you do a control study that included other individuals answering the exact same questions as the breeders to see if this is breeder specific, area specific, or not related? If so, please mention this. This can only add to your results.

Author Response

Reviewer #2.

Figueroa et al. created the Gamefowl Wellness Program in response to the lack of education surrounding the 2018 Newcastle disease virus outbreak and prevention measures in California. Through a primary needs assessment, focus groups with breeders, and outreach with feedstores, the authors were able to determine the resources this community needed and in which formats they best responded to. This has allowed for preemptive work to be done in the communities that continue to see virulent Newcastle disease virus outbreaks.

In my opinion, this work has shown the gap between research completed in the laboratory and dissemination of that information into the community. Other projects like this one have been completed for other viral infections, but to my knowledge, this is the first for NDV, following the latest outbreak. The novelty and quality of this manuscript, along with the interest being there for a wide audience, I suggest that this article be accepted with minor revisions.

We appreciate the reviewer comment  

I do have a few minor remarks:

  1. Please post the links in the results section for the podcast and the facebook group, as these were part of the data for the article.

The links were added in the materials and methods section of the manuscript

  1. Since questions are a part of the results for this study, a table may show this data in a format that is easier for others to replicate the study.

Table 1 was created to show the poll questions and its answers

  1. Error bars need to be inserted in each figure that has a bar graph.

Since the poll now is represented by the table, graphs were eliminated.  

  1. Each graph should equal 100%. In one figure it equals 99%, one figure equals 101%, another 107%, and another 120%. Exact numbers (n=) would be useful in the results section since in some instances you only have 5 participants. Maybe with the table of questions?

A table replaced the graphs

  1. Did you have any control questions for this experiment? If so, please mention them. If not, this may be helpful in the future.

No control questions were designed this time, we will consider the reviewer suggestion  

  1. Did you do a control study that included other individuals answering the exact same questions as the breeders to see if this is breeder specific, area specific, or not related? If so, please mention this. This can only add to your results.

We appreciate the reviewer feedback, unfortunately we did not use a control population.

Round 2

Reviewer 1 Report

General Comments:

The authors did a really nice job of changing the tone of the manuscript and addressing previous concerns. A few small edits are highlighted below.

Line 70 – change to owners of flock at risk – it’s the owners who are resistant

Line 96 – add an “and” before a strict protocol

Line 197 – should this be interactions instead of intercations?

Line 207 – not sure what is missing here, but the sentence reads strangely

Line 220 – misspelling of preferred

Line 227 – suggest rewording – creating the need to brainstorm

Line 328 – remove the word constantly – while there was evidence that some gamefowl were transported within and outside of the quarantine area resulting in disease spread, it was not constant. The State Veterinarian’s efforts to control movement were critical to the overall success of the response and did have an impact on reducing movements.

Line 372 – misspelling of improperly

Author Response

Reviewer #1.

The authors did a really nice job of changing the tone of the manuscript and addressing previous concerns. A few small edits are highlighted below.

We are glad to hear that we met your expectations

Line 70 – change to owners of flock at risk – it’s the owners who are resistant

This was changed

Line 96 – add an “and” before a strict protocol

“and” was added

Line 197 – should this be interactions instead of intercations?

This was fixed

Line 207 – not sure what is missing here, but the sentence reads strangely

The sentence was rephrased for clarity

Line 220 – misspelling of preferred

This was corrected

Line 227 – suggest rewording – creating the need to brainstorm

This was reviewed

Line 328 – remove the word constantly – while there was evidence that some gamefowl were transported within and outside of the quarantine area resulting in disease spread, it was not constant. The State Veterinarian’s efforts to control movement were critical to the overall success of the response and did have an impact on reducing movements.

The word “constantly” was removed

Line 372 – misspelling of improperly

This was corrected